# Detecting images generated by diffusers

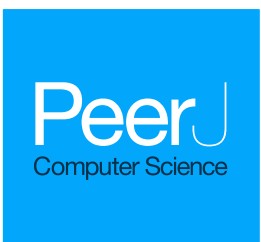

Davide Alessandro Coccomini[1,2], Andrea Esuli[1], Fabrizio Falchi[1], Claudio Gennaro[1] and Giuseppe Amato[1]

[1] Institute of Information Science and Technologies "Alessandro Faedo", Italian National Research Council, Pisa, Tuscany, Italy
[2] Information Engineering, University of Pisa, Pisa, Tuscany, Italy

## ABSTRACT

In recent years, the field of artificial intelligence has witnessed a remarkable surge in the generation of synthetic images, driven by advancements in deep learning techniques. These synthetic images, often created through complex algorithms, closely mimic real photographs, blurring the lines between reality and artificiality. This proliferation of synthetic visuals presents a pressing challenge: how to accurately and reliably distinguish between genuine and generated images. This article, in particular, explores the task of detecting images generated by text-to-image diffusion models, highlighting the challenges and peculiarities of this field. To evaluate this, we consider images generated from captions in the MSCOCO and Wikimedia datasets using two state-of-the-art models: Stable Diffusion and GLIDE. Our experiments show that it is possible to detect the generated images using simple multi-layer perceptrons (MLPs), starting from features extracted by CLIP or RoBERTa, or using traditional convolutional neural networks (CNNs). These latter models achieve remarkable performances in particular when pretrained on large datasets. We also observe that models trained on images generated by Stable Diffusion can occasionally detect images generated by GLIDE, but only on the MSCOCO dataset. However, the reverse is not true. Lastly, we find that incorporating the associated textual information with the images in some cases can lead to a better generalization capability, especially if textual features are closely related to visual ones. We also discovered that the type of subject depicted in the image can significantly impact performance. This work provides insights into the feasibility of detecting generated images and has implications for security and privacy concerns in real-world applications. The code to reproduce our results is available at: https://github.com/davide-coccomini/Detecting-Images-Generated-by-Diffusers.

## INTRODUCTION

The rapid progression of synthetic image generation techniques, notably through the advent of generative adversarial networks (GANs) and diffusion models, has ushered in an era where artificial images closely resemble their real counterparts. While these advancements hold immense potential for creative applications, they have also raised significant concerns regarding their misuse, particularly in the form of deepfakes and the

Corresponding author
Davide Alessandro Coccomini,
davidealessandro.coccomini@isti.cnr.it

spread of misinformation. These manipulated contents can indeed be used to distort reality, damage the reputation of people, and irretrievably ruin their lives.

To mitigate these concerns, it is crucial to develop robust techniques for detecting synthetic images and distinguishing them from pristine content. The ability to detect synthetic images is essential for maintaining the integrity of information and for protecting individuals from the malicious use of these manipulated media. The explosion of recent text-to-image methods such as Stable Diffusion (*Rombach et al., 2022*) and their easy access to the general public is leading society towards a point where a good deal of online content is synthetic and where the line between reality and fiction will no longer be so clear.

In this article, we present blackan analysis of peculiarities and challenges in the field of synthetic image detection, considering classifications based on the image itself and the text associated with it and used to describe or generate it. To explore the field of synthetic image detection, we constructed two datasets based on different sets of captions and pristine images. We consider two generators, namely Stable Diffusion and GLIDE to construct these datasets. We subsequently trained several binary classifiers based on various deep-learning architectures, also introducing some multimodal strategies exploiting features extracted from both images and associated texts. We also analyzed the peculiarities of images, such as the kind of object depicted in the scene, and texts, such as linguistic features, that can lead to a more or less credible image that is difficult to identify. Throughout this analysis, we have been able to highlight which conditions can lead to a better synthetic image and which ones do not influence the generation process. In general, we try to answer the following research questions:

- Is there a generalization problem similar to what we see in the traditional deepfake detection field?
- What is the impact of models' pretraining on synthetic image detection?
- Can the combination of visual and textual features conduct an improvement in deepfake detection?
- What are the specific visual and textual characteristics that contribute to the credibility of synthetic images?

Portions of this text were previously published as part of a preprint (*Coccomini et al., 2023b*).

## RELATED WORKS

### Images generation

One of the earliest GAN-based methods for synthetic image generation was introduced in *Goodfellow et al. (2014)*. Their model consisted of a generator network that synthesized images and a discriminator network that learned to distinguish between real and synthetic images. They have been used for several tasks like face synthesis (*Ruiz et al., 2020*; *Mokhayeri, Kamali & Granger, 2019*), style transfer (*Xu et al., 2021*) and super-resolution (*Ledig et al., 2017*). Impressive results have been achieved in *Karras et al. (2020)* where the

authors proposed style-based GAN architecture (StyleGAN), capable of generating more credible images. Several variations of GAN architecture have been designed over the years, for example, the CycleGAN proposed in *Zhu et al. (2017)*. This architecture can perform the task of image-to-image translation, optimizing a cycle loss and also allowing the revert of the mapping process. GAN architecture has also been used for the task of text-to-image translation in which a text describing a context is converted into an image. In particular, in *Qiao et al. (2019)* the MirrorGAN is proposed. The authors exploit the concept of re-description in the sense that the image generated starting from a text, should be describable by another text which is similar to the source one. So, the model needs to learn to generate images whose re-description matches as much as possible with the requested one. Recently a novel architecture was introduced to perform similar tasks, namely diffusion models. These models generate images by refining an initial noise vector through multiple diffusion steps. For text-to-image generation, a given text is encoded into a latent vector used as the initial noise. Some of these models obtained unprecedented results such as DALL-E (*Ramesh et al., 2021*), GLIDE (*Nichol et al., 2022*) and Stable Diffusion (*Rombach et al., 2022*). These models have a high control of image generation, allowing the users to create very credible images with a high level of detail.

## Syntethic images detection

The growing credibility and diffusion of generated images raised some concerns in the research community, which tried to develop methods capable of effectively distinguishing synthetic content from pristine one. For a long time, the main efforts were focused on traditional deepfakes referring to face manipulation, with a lot of works proposed trying to detect them (*Coccomini et al., 2022c*; *Zheng et al., 2021*; *Coccomini et al., 2022a*; *Guarnera et al., 2022*; *Coccomini et al., 2023a*; *Baxevanakis et al., 2022*; *Li et al., 2020*; *Coccomini et al., 2022b*; *Caldelli et al., 2021*; *Zhang et al., 2022*; *Cozzolino et al., 2020*). Otherwise, with the recent advancement of generation techniques, attention has also been posed to generic-generated content without a fundamental focus on images depicting humans. For example, in *Sha et al. (2022)* the authors tried to detect images generated by some diffusion models in two setups, image-only and image+text being capable of effectively distinguishing between real and generated images. In *Corvi et al. (2022)*, the researchers tried to train some binary classifiers to distinguish images generated by diffusion models and GAN models. The results highlighted how it is pretty feasible to detect images when the generated method is used in the training set while there is a huge generalization problem. blackThis behaviour resides in what was also observed by *Bammey (2024)* which confirmed the presence of specific artifacts for each diffusion model introduced in the frequency domain. These are memorized by detectors that can be very effective in training methods but little in classifying images generated by novel techniques. The specific attributes introduced by diffusion-based generators have been also exploited in *Ma et al. (2023)* where the authors proposed SeDID, an effective combination of statistical and neural network-based approach to catch them. Indeed, the classifiers seem to learn some kind of trace specific for each generator, and so are pretty limited when tested on images generated with other methods. Similar work has recently been done in *Amoroso et al.*

*(2023)* presenting a novel synthetic dataset, namely COCOFake, and highlighting the presence of common low-level cues in images generated by state-of-the-art diffusion models. Although many previous works have explored the problem of synthetic image detection, in this article we aim to conduct a deeper analysis of the impact of the various aspects that influence this task. In particular, we will explore the role of the subjects depicted in the images on a more or less credible image as well as the influence of the linguistic structure of the captions associated with them. We will then analyze the impact of the architecture choice and the role of pretraining and different feature extraction backbones, trying to highlight their peculiarities, advantages and disadvantages.

## EXPERIMENTS

In this section, we explain all the experiments conducted and the details to reproduce them.

### Classifiers

As presented in Fig. 1, we conducted our experiments in two main setups: image-only and text+image. In the first case, classification is done by using only the features extracted from the image. For this purpose, we selected some simple deep learning architectures to use as binary classifiers (real or generated images). In particular, the first model used is a simple MLP that takes as input the features extracted from the image by CLIP (*Radford et al., 2021*) encoders. Contrastive Language-Image Pre-Training (CLIP) is a neural network model that learns to associate natural language descriptions with images. For that reason it extracts very correlated textual and visual features. CLIP can use both ResNet50 (*He et al., 2016*) and Vision Transformer (ViT) (*Dosovitskiy et al., 2020*) features to represent the images it processes. The second category of trained models is standard convolutional neural networks (CNN) widely used in computer vision *i.e.*, Resnet50 and XceptionNet (*Chollet, 2017*), both pretrained for image classification on the ImageNet dataset (*Deng et al., 2009*).

We extended the CLIP-based classifier from the previous setup by also using the text encoder provided by the architecture to extract features from the caption associated with the image. These textual features are combined with the visual features and given as input to the MLP for classification. Some experiments have been also done using the textual features extracted using a RoBERTa model (*Liu et al., 2019*), replacing the CLIP text encoder. RoBERTa is a pre-trained natural language processing model based on transformer architecture, designed to understand and generate human-like text by learning contextual relationships in large datasets. The features extracted with this method could be richer in linguistic information compared with CLIP textual features due to the different nature of the task the model is trained on. On the other hand, the CLIP textual features should be easier to correlate with visual features to find inconsistencies between the caption and the image. The visual features remain extracted through the CLIP model also when the RoBERTa text-encoder is exploited.

In both setups, the MLP structure is adapted to the features' shape based on the model used for the extraction, while the other layers remain the same for all the experiments. In

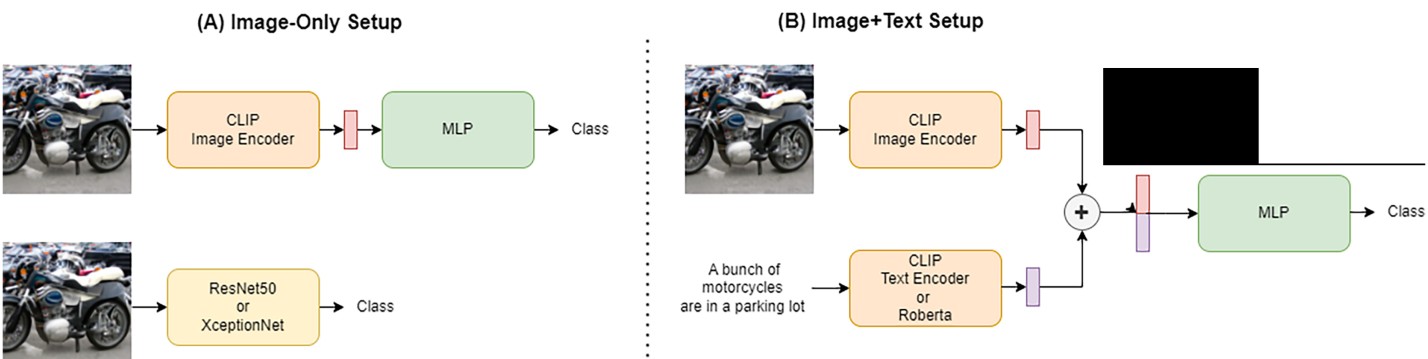

**Figure 1** **Figure showing the two training setup.** The two different network setups are shown in the figure. (A) The Image-only setup in which only the image under consideration is used as input to the network while ignoring its associated caption. This setup is used for some MLPs and the two convolutional networks under consideration. (B) On the other hand, the Image+text setup is presented in which the features obtained from the image are concatenated with those obtained from the caption, both extracted *via* CLIP or using a RoBERTa model for the textual features and given as input to the network.

particular, the model is composed of one input layer, two hidden layers, and one output layer. All models used in the experiments have a similar number of parameters to allow for a fair comparison.

## Dataset

To validate the ability of a classifier to effectively identify images generated from text, we considered two starting datasets, namely MSCOCO (*Lin et al., 2014*) composed by over 330,000 English captioned images and Wikimedia Image-Caption Matching dataset (https://www.kaggle.com/c/wikipedia-image-caption/overview) based on Wikipedia Image Text Dataset (WIT) (*Srinivasan et al., 2021*) and contains 37.6 million entity-rich image-text examples with 11.5 million unique images across 108 Wikipedia languages. Both of these datasets consist of images from many different contexts and each of them is associated with a caption describing it as shown in Fig. 2. For each of the two datasets, 6,000 images randomly sampled from the training set, and a further 6,000 images were generated using two text-to-image methods, namely Stable Diffusion v1.4 or GLIDE. Therefore, in the constructed sets, for each caption sampled from the source dataset there is one pristine and one fake image generated using a text-to-image model. The same was done with 1,500 images from the validation set and another 6,000 from the test set. Therefore, all models were trained on a training set consisting of 12,000 images, half of which were generated by Stable Diffusion, and the same have been done with another training set with images generated by GLIDE. We then constructed four different sets, each one composed of a total of 27,000 images, split in train/validation/test based on the annotations provided by the datasets' authors. To perform a more detailed analysis of the behaviour of the classifiers considered, the images from the Wikimedia dataset and the corresponding generated images were categorized based on the Wikipedia ontologies available online. An overview of the constructed datasets is shown in Table 1 and the code to reproduce them is available at *Coccomini et al. (2024)*.

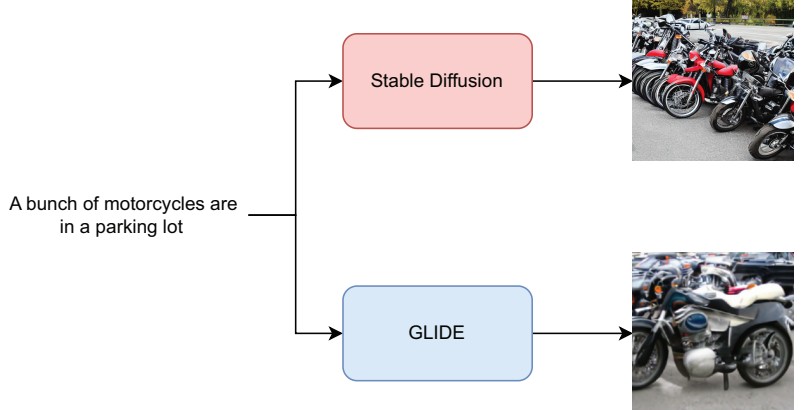

**Figure 2 Figure showing the image generation process.** An example of a caption associated with an image from the MSCOCO dataset given as input to Stable Diffusion and GLIDE to generate two additional images is shown in the figure.

**Table 1 Summary of the constructed datasets indicating the source of pristine images and the technique of image generation starting from their captions.** The "categorized" column indicates if information about the category of the images is available.

|       | Pristine source | Generation method | Categorized |
|-------|-----------------|-------------------|-------------|
| Set 1 | MSCOCO          | Stable Diffusion  | ×           |
| Set 2 | MSCOCO          | GLIDE             | ×           |
| Set 3 | Wikimedia       | Stable Diffusion  | ✓           |
| Set 4 | Wikimedia       | GLIDE             | ✓           |

## Training setup

The classifiers considered were simple multi-layer perceptrons (MLPs), which take visual and textual features from CLIP, RoBERTa or widely used convolutional networks, in particular, ResNet50 and XceptionNet. The former are trained from scratch, while the convolutional networks are considered also in the case of ImageNet pretraining. All models are trained with a learning rate of 0.1, decreasing to 0.001, for up to 270 epochs on an NVIDIA A100. MLPs are trained in two possible setups: image-only and text+image. CLIP-extracted features are extracted from the image in both the setups while in the text +image context, the captions' features are obtained through CLIP text-encoder or using RoBERTa. In the text+image setup, the features are concatenated before being given as input to the model as shown in Fig. 1, which is then able to see both the textual and visual components in a single vector. The encoders used in all these experiments are frozen and just used for feature extraction. Instead, the CNNs considered are trained exclusively in image-only mode, and the features used for classification are those resulting from the convolutional layers of the considered architecture. To explore the possibility of training also the backbone, in this case, we fine-tuned the whole architecture.

**Table 2 Results on the test set of the various classifiers trained and tested on real images and images generated with Stable Diffusion.** For each row, the dataset considered, training mode used, features extracted *via* CLIP or convolutional layers, accuracy and AUC, number of model parameters, and whether pretraining was used are indicated. The bold numbers indicate the best result obtained by the models on a specific dataset.

| Model | Dataset | Mode | Features | Accuracy ↑ | AUC ↑ | Params | Pretrain |
|---|---|---|---|---|---|---|---|
| MLP-Base | MSCOCO | Image Only | CLIP-VIT | 79.5 | 88.8 | 23M | N/A |
| MLP-Base | MSCOCO | Text+Image | CLIP-VIT | 78.5 | 88.8 | 23M | N/A |
| MLP-RoBERTa | MSCOCO | Text+Image | CLIP-VIT | 75.4 | 89.1 | 23M | N/A |
| MLP-Base | MSCOCO | Image Only | CLIP-R50 | 67.5 | 75.0 | 23M | N/A |
| MLP-Base | MSCOCO | Text+Image | CLIP-R50 | 66.5 | 74.2 | 23M | N/A |
| MLP-RoBERTa | MSCOCO | Text+Image | CLIP-R50 | 59.3 | 75.0 | 23M | N/A |
| XceptionNet | MSCOCO | Image Only | XceptionNet | 90.6 | 96.6 | 20M | N/A |
| XceptionNet | MSCOCO | Image Only | XceptionNet | 94.6 | 98.9 | 20M | ImageNet |
| Resnet50 | MSCOCO | Image Only | Resnet50 | 87.8 | 94.3 | 23M | N/A |
| Resnet50 | MSCOCO | Image Only | Resnet50 | **97.1** | **99.6** | 23M | ImageNet |
| MLP-Base | Wikipedia | Image Only | CLIP-VIT | 72.8 | 81.4 | 23M | N/A |
| MLP-Base | Wikipedia | Text+Image | CLIP-VIT | 73.1 | 80.8 | 23M | N/A |
| MLP-RoBERTa | Wikipedia | Text+Image | CLIP-VIT | 58.1 | 81.6 | 23M | N/A |
| MLP-Base | Wikipedia | Image Only | CLIP-R50 | 65.9 | 74.2 | 23M | N/A |
| MLP-Base | Wikipedia | Text+Image | CLIP-R50 | 64.5 | 73.5 | 23M | N/A |
| MLP-RoBERTa | Wikipedia | Text+Image | CLIP-R50 | 67.7 | 74.5 | 23M | N/A |
| XceptionNet | Wikipedia | Image Only | XceptionNet | 82.3 | 91.2 | 20M | N/A |
| XceptionNet | Wikipedia | Image Only | XceptionNet | 90.7 | 97.1 | 20M | ImageNet |
| Resnet50 | Wikipedia | Image Only | Resnet50 | 78.1 | 87.8 | 23M | N/A |
| Resnet50 | Wikipedia | Image Only | Resnet50 | **94.5** | **98.1** | 23M | ImageNet |

# RESULTS

In this section, we show the results obtained in two main contexts, intra-method and cross-method.

## In-method classifications

Table 2 shows the performances, in terms of accuracy and AUC, of the classifiers considered when trained and tested with real images and images generated by Stable Diffusion, according to the setup illustrated above. As can be seen from the results, the pretrained Resnet50 and XceptionNet exhibit remarkable capabilities in identifying generated images by acting as almost perfect detectors. However, the same models demonstrate lower accuracy in classification in the absence of pretraining, although remaining vastly more effective than MLP-based classifiers. Indeed, on the other hand, MLPs achieve satisfactory results especially when CLIP visual features extracted *via* a Vision Transformer are used, without the use of pretrain on large datasets. Another element that greatly influences the result, especially for MLPs is certainly the dataset, images from Wikimedia seem to be more difficult to distinguish. This probably results

from the wide variety of images and contexts in this dataset, however, even in this case the models manage to achieve good levels of accuracy.

As shown instead in Table 3, the images generated *via* GLIDE seem to be much easier to detect on both datasets, even without the use of pretraining. The MLP with CLIP-VIT features is by far the best MLP setup. Pretrained XceptionNet and Resnet50 perform almost perfectly with accuracies around 99%. The absence of pretraining in this case is more impactful on the Resnet50 than on the XceptionNet. The latter scores a very good performance even without pretraining probably exploiting some particular artifacts introduced by GLIDE. As can be seen in Fig. 3, the images generated by GLIDE appear to be more artefactual and bogus than those generated by Stable Diffusion from the same caption, which obtains more credible results. Just as the latter are more difficult to identify with the naked eye, the models experience similar difficulties.

In many experiments, using RoBERTa's textual features instead of the ones extracted through the CLIP text encoder led to decreased performances. This means that the textual features extracted using the CLIP encoder are more expressive and clear in combination with the visual ones allowing the classifier to reach higher performances in the majority of setups.

## Cross-method classification

The trained models have shown a great ability to identify images generated by text-to-image systems. However, in this section, we want to find out whether they can generalize the concept of generated images to such an extent that they can identify images created with different generators. Indeed, many text-to-image systems are available, and a real-world deployed model should be able to identify generated images regardless of the method used. The risk is in fact that these classifiers learn to distinguish some sort of trace, noise or imprint left by the generator instead of focusing on more general, high-level anomalies or inconsistencies, thus rendering them useless in the real world. To validate this, we tried training models on real images and images generated with Stable Diffusion and tested them on images generated with GLIDE, and *vice versa*.

In Table 4 we report the results of the same models evaluated in "In-Method Classification" in a cross-forgery context. The accuracy figures in this more realistic setup are sensibly lower and the MLP models using ViT features close the gap concerning CNNs. Two models stand out in terms of accuracy and AUC, namely the MLP trained in the image+text setup using CLIP textual features and the Resnet50. The latter succeeds in generalizing better than other models but only when pretrained, highlighting how pretraining models on large datasets can be of crucial help in improving generalization. The MLP-based detector trained on the image+text setup has a substantial positive difference from the same model trained in the image-only one. Since the model can no longer fully exploit visual information, not having had the opportunity to learn artifacts from images generated by other methods, it exploits the linguistic information contained in the caption. In particular, the model could notice the inconsistencies between what is described in the caption and what is represented in the images. In fact, both the language features extracted using RoBERTa and those obtained through CLIP lead to improved

**Table 3 Results on the test set of the various classifiers trained and tested on real images and images generated with GLIDE.** For each row, the dataset considered, training mode used, features extracted *via* CLIP or convolutional layers, accuracy and AUC, number of model parameters, and whether pretraining was used are indicated. The bold numbers indicate the best result obtained by the models on a specific dataset.

| Model | Dataset | Mode | Features | Accuracy ↑ | AUC ↑ | Params | Pretrain |
|---|---|---|---|---|---|---|---|
| MLP-Base | MSCOCO | Image only | CLIP-VIT | 95.8 | 99.2 | 23M | N/A |
| MLP-Base | MSCOCO | Text+Image | CLIP-VIT | 95.8 | 99.2 | 23M | N/A |
| MLP-RoBERTa | MSCOCO | Text+Image | CLIP-VIT | 96.3 | 99.2 | 23M | N/A |
| MLP-Base | MSCOCO | Image only | CLIP-R50 | 79.0 | 87.4 | 23M | N/A |
| MLP-Base | MSCOCO | Text+Image | CLIP-R50 | 78.0 | 86.4 | 23M | N/A |
| MLP-RoBERTa | MSCOCO | Text+Image | CLIP-R50 | 77.4 | 87.1 | 23M | N/A |
| XceptionNet | MSCOCO | Image only | XceptionNet | 99.3 | 99.9 | 20M | N/A |
| XceptionNet | MSCOCO | Image only | XceptionNet | 99.2 | 99.9 | 20M | ImageNet |
| Resnet50 | MSCOCO | Image only | Resnet50 | 80.0 | 93.1 | 23M | N/A |
| Resnet50 | MSCOCO | Image only | Resnet50 | **99.3** | **99.9** | 23M | ImageNet |
| MLP-Base | Wikipedia | Image only | CLIP-VIT | 93.7 | 98.4 | 23M | N/A |
| MLP-Base | Wikipedia | Text+Image | CLIP-VIT | 94.3 | 98.4 | 23M | N/A |
| MLP-RoBERTa | Wikipedia | Text+Image | CLIP-VIT | 93.5 | 98.4 | 23M | N/A |
| MLP-Base | Wikipedia | Image only | CLIP-R50 | 77.1 | 85.2 | 23M | N/A |
| MLP-Base | Wikipedia | Text+Image | CLIP-R50 | 75.5 | 84.5 | 23M | N/A |
| MLP-RoBERTa | Wikipedia | Text+Image | CLIP-R50 | 76.4 | 84.8 | 23M | N/A |
| XceptionNet | Wikipedia | Image only | XceptionNet | 98.7 | 99.8 | 20M | N/A |
| XceptionNet | Wikipedia | Image only | XceptionNet | 98.9 | 99.9 | 20M | ImageNet |
| Resnet50 | Wikipedia | Image only | Resnet50 | 74.3 | 88.4 | 23M | N/A |
| Resnet50 | Wikipedia | Image only | Resnet50 | **99.5** | **99.9** | 23M | ImageNet |

2 giraffes standing under a tree in the shade.

A airplane that is on a runway by some grass.

A bathroom with a sink, standup shower and tub.

GLIDE

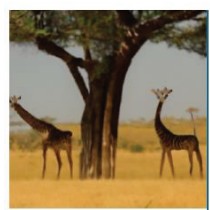
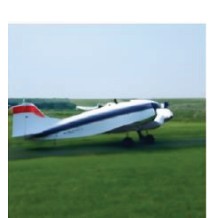
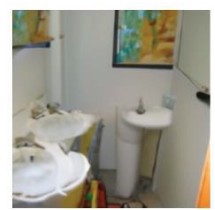

Stable Diffusion

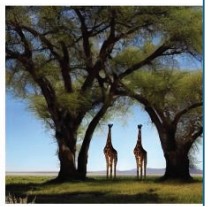
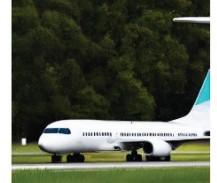
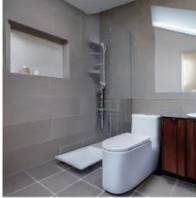

**Figure 3 Images generated with GLIDE and Stable Diffusion.** Comparison between GLIDE and Stable Diffusion generated images from the same caption.

**Table 4 Results obtained by the considered classifiers trained on real images and generated with Stable Diffusion and then tested on real images and generated with GLIDE, and *vice versa* on the MSCOCO dataset.** The bold numbers indicate the best result obtained by the models on a specific combination of Training and Testing method.

| Model | Training method | Testing method | Mode | Features | Accuracy ↑ | AUC ↑ | Pretrain |
|---|---|---|---|---|---|---|---|
| MLP-Base | Stable Diffusion | GLIDE | Image–Only | CLIP-R50 | 50.8 | 50.3 | N/A |
| MLP-Base | Stable Diffusion | GLIDE | Image+Text | CLIP-R50 | 49.9 | 50.9 | N/A |
| MLP-Roberta | Stable Diffusion | GLIDE | Image+Text | CLIP-R50 | 50.4 | 50.8 | N/A |
| MLP-Base | Stable Diffusion | GLIDE | Image–Only | CLIP-VIT | 64.9 | 75.7 | N/A |
| MLP-Base | Stable Diffusion | GLIDE | Image+Text | CLIP-VIT | **69.7** | 76.2 | N/A |
| MLP-Roberta | Stable Diffusion | GLIDE | Image+Text | CLIP-VIT | 60.9 | 75.3 | N/A |
| XceptionNet | Stable Diffusion | GLIDE | Image–Only | XceptionNet | 53.1 | 57.6 | N/A |
| XceptionNet | Stable Diffusion | GLIDE | Image–Only | XceptionNet | 58.9 | 76.1 | ImageNet |
| Resnet50 | Stable Diffusion | GLIDE | Image–Only | Resnet50 | 53.8 | 59.3 | N/A |
| Resnet50 | Stable Diffusion | GLIDE | Image–Only | Resnet50 | 61.4 | **86.1** | ImageNet |
| MLP-Base | GLIDE | Stable Diffusion | Image–Only | CLIP-R50 | 51.5 | 51.7 | N/A |
| MLP-Base | GLIDE | Stable Diffusion | Image+Text | CLIP-R50 | 50.6 | 51.1 | N/A |
| MLP-Roberta | GLIDE | Stable Diffusion | Image+Text | CLIP-R50 | 51.4 | 51.7 | N/A |
| MLP-Base | GLIDE | Stable Diffusion | Image–Only | CLIP-VIT | 48.8 | 45.9 | N/A |
| MLP-Base | GLIDE | Stable Diffusion | Image+Text | CLIP-VIT | **52.3** | 61.5 | N/A |
| MLP-Roberta | GLIDE | Stable Diffusion | Image+Text | CLIP-VIT | 51.1 | 60.9 | N/A |
| XceptionNet | GLIDE | Stable Diffusion | Image–Only | XceptionNet | 50.0 | 67.5 | N/A |
| XceptionNet | GLIDE | Stable Diffusion | Image–Only | XceptionNet | 50.2 | **73.5** | ImageNet |
| Resnet50 | GLIDE | Stable Diffusion | Image–Only | Resnet50 | 48.9 | 47.7 | N/A |
| Resnet50 | GLIDE | Stable Diffusion | Image–Only | Resnet50 | 50.2 | 64.6 | ImageNet |

performance. The latter, however, are even more effective because they are designed to relate what is described in the caption and what is represented in the image, thus making it easier for the model to find inconsistencies. A similar trend can be observed in the case of models trained with images generated using GLIDE and tested with images obtained through Stable Diffusion. In this case, the classification is more complex, in fact the images generated with Stable Diffusion are more challenging and training on GLIDE-generated images is not enough for the model to detect them. Nevertheless, again the introduction of textual features leads to improved performance.

The same experiments were conducted on Wikimedia datasets as illustrated in Table 5. In this context, the difficulties that the models had already encountered in the previous experiments become even more evident and stem probably from the greater challenge of the dataset. As it is much more varied than MSCOCO, it makes generalization more difficult, as the models have to learn to distinguish between images generated from very varied contexts. Practically no model manages to exceed the accuracy of 50%, demonstrating a total inability to generalize. The introduction of textual features also does not help to improve classification, and this can also be related to the different nature of captions compared to those in MSCOCO. In fact, from our observations, Wikimedia's

**Table 5 Results obtained by the considered classifiers trained on real images and generated with Stable Diffusion and then tested on real images and generated with GLIDE, and *vice versa* on the Wikimedia dataset.** The bold numbers indicate the best result obtained by the models on a specific combination of Training and Testing method.

| Model | Training method | Testing method | Mode | Features | Accuracy ↑ | AUC ↑ | Pretrain |
|---|---|---|---|---|---|---|---|
| MLP-Base | Stable Diffusion | GLIDE | Image–Only | CLIP-R50 | 36.4 | 44.8 | N/A |
| MLP-Base | Stable Diffusion | GLIDE | Image+Text | CLIP-R50 | 36.7 | 45.7 | N/A |
| MLP-Roberta | Stable Diffusion | GLIDE | Image+Text | CLIP-R50 | 41.0 | 36.2 | N/A |
| MLP-Base | Stable Diffusion | GLIDE | Image–Only | CLIP-VIT | 46.2 | 40.5 | N/A |
| MLP-Base | Stable Diffusion | GLIDE | Image+Text | CLIP-VIT | 43.4 | 34.4 | N/A |
| MLP-Roberta | Stable Diffusion | GLIDE | Image+Text | CLIP-VIT | 49.1 | 38.2 | N/A |
| XceptionNet | Stable Diffusion | GLIDE | Image–Only | XceptionNet | 46.2 | 37.7 | N/A |
| XceptionNet | Stable Diffusion | GLIDE | Image–Only | XceptionNet | 48.9 | 48.8 | ImageNet |
| Resnet50 | Stable Diffusion | GLIDE | Image–Only | Resnet50 | 47.6 | 42.8 | N/A |
| Resnet50 | Stable Diffusion | GLIDE | Image–Only | Resnet50 | **50.9** | **55.0** | ImageNet |
| MLP-Base | GLIDE | Stable Diffusion | Image–Only | CLIP-R50 | 45.6 | 39.9 | N/A |
| MLP-Base | GLIDE | Stable Diffusion | Image+Text | CLIP-R50 | 39.4 | 45.8 | N/A |
| MLP-Roberta | GLIDE | Stable Diffusion | Image+Text | CLIP-R50 | 45.8 | 39.8 | N/A |
| MLP-Base | GLIDE | Stable Diffusion | Image–Only | CLIP-VIT | 48.8 | 45.9 | N/A |
| MLP-Base | GLIDE | Stable Diffusion | Image+Text | CLIP-VIT | 48.0 | 45.8 | N/A |
| MLP-Roberta | GLIDE | Stable Diffusion | Image+Text | CLIP-VIT | 48.8 | 45.6 | N/A |
| XceptionNet | GLIDE | Stable Diffusion | Image–Only | XceptionNet | 50.9 | 49.6 | N/A |
| XceptionNet | GLIDE | Stable Diffusion | Image–Only | XceptionNet | **49.3** | 49.6 | ImageNet |
| Resnet50 | GLIDE | Stable Diffusion | Image–Only | Resnet50 | 42.7 | 39.7 | N/A |
| Resnet50 | GLIDE | Stable Diffusion | Image–Only | Resnet50 | 49.2 | **53.7** | ImageNet |

captions can be much less descriptive, so finding inconsistencies between text and image can be more complex.

## Error analysis

The structure of the Wikimedia dataset allows us to perform an error analysis by category. As mentioned above, we have predictively categorized the images in this dataset into various categories (*e.g.*, Artist, City, Road, River, Animal, *etc.*,) based on the ontologies provided by Wikipedia and the tags associated with each image. They were then further grouped into two macro-categories, namely inanimate and animate objects. Based on these categories, we analyzed the errors committed by the classifiers on the test set.

In Table 6, we give an example of the percentage false negative, namely undetected generated images, and percentage false positive, namely wrongly classified real images, made on the Wikimedia test set of trained models. Looking at the false negative results, the models tend to have more difficulty identifying generated images when the image depicts an inanimate object (*e.g.*, buildings, roads, objects, infrastructure, rivers *etc.*,) than when it is an image with animate subjects (people, animals *etc.*,). In other words, images generated by both models depicting inanimate subjects are more believable and, therefore, more difficult to distinguish from real images. For example, generating a credible person or

**Table 6 Percentage of false negatives and false positives in the two categories considered, animate and inanimate objects, on Wikimedia test dataset with images generated with Stable Diffusion or GLIDE.**

| Model | Mode | Features | Stable Diffusion | | | | GLIDE | | | | Pretrain |
| --- | --- | --- | --- | --- | --- | --- | --- | --- | --- | --- | --- |
| | | | Animated | | Inanimate | | Animated | | Inanimate | | |
| | | | FN↓ | FP↓ | FN↓ | FP↓ | FN↓ | FP↓ | FN↓ | FP↓ | |
| MLP-Base | Image–Only | CLIP-R50 | 26.4 | 8.3 | 31.0 | 16.4 | 15.9 | 8.3 | 20.6 | 16.1 | N/A |
| MLP-Base | Image+Text | CLIP-R50 | 28.6 | 7.6 | 35.0 | 15.7 | 17.6 | 7.5 | 23.3 | 15.7 | N/A |
| MLP-Roberta | Image+Text | CLIP-R50 | 12.6 | 34.0 | 16.6 | 31.0 | 16.7 | 8.4 | 21.4 | 15.8 | N/A |
| MLP-Base | Image–Only | CLIP-VIT | 21.7 | 14.3 | 25.0 | 16.6 | 5.6 | 3.0 | 7.0 | 5.0 | N/A |
| MLP-Base | Image+Text | CLIP-VIT | 17.6 | 19.4 | 21.0 | 18.7 | 2.6 | 6.6 | 3.8 | 7.6 | N/A |
| MLP-Roberta | Image+Text | CLIP-VIT | 45.2 | 1.4 | 45.7 | 2.4 | 5.8 | 3.0 | 7.0 | 4.7 | N/A |
| XceptionNet | Image–Only | XceptionNet | 16.7 | 11.3 | 19.4 | 10.7 | 1.3 | 0.3 | 1.3 | 1.3 | N/A |
| XceptionNet | Image–Only | XceptionNet | 6.0 | 7.9 | 7.4 | 7.0 | 1.1 | 0.1 | 2.8 | 1.0 | ImageNet |
| Resnet50 | Image–Only | Resnet50 | 17.8 | 11.0 | 24.8 | 13.4 | 4.3 | 33.6 | 4.3 | 34.1 | N/A |
| Resnet50 | Image–Only | Resnet50 | 6.0 | 3.0 | 5.8 | 4.0 | 0.2 | 0.2 | 1.1 | 1.0 | ImageNet |

animal in the eyes of the classifier thus seems to be a more difficult task for the text-to-image systems under consideration. Some structures in the human body seem to be particularly challenging to generate such as the hand's fingers, increasing the probability of artifact introduction. On the other hand, an inanimate object due to its variety, can contain anomalies or inconsistencies introduced by generators that could easily be mistaken for normality or peculiarities of the scene.

## Linguistic analysis

To better understand the nature of the errors made by the classifiers, we investigated linguistically the captioning associated with them in an attempt to understand whether the patterns are influenced by particular elements in the sentences. The correlation between the correct or incorrect classification of an image and the linguistic features listed and described in Table 7 was analyzed. These linguistic features are extracted using a language processing pipeline trained on blogs and web articles taken from SpaCy (https://spacy.io/). This pipeline can analyze sentences and extract linguistic features from them, such as if a token is a conjunction, an adjective and so on.

Figure 4 shows examples of Pearson correlation between the classification of an MLP trained on real images and images generated by one of the two methods analyzed (Stable Diffusion and GLIDE) and the linguistic features considered. To do so, these were extracted from each caption associated with the test images, and the correlation between their variation and the classification provided by the model was derived. As can be seen from the figure, there is no strong correlation between these features and the classification. It thus appears to be more influenced by the category of the image rather than the composition of the sentence.

**Table 7 Features used for the linguistic analysis conducted with their acronyms.**

| Feature | Description |
|---|---|
| LENGTH | The length of the caption |
| ADJ | Number of adjectives (*e.g.*, big, old, green) |
| ADP | Number of prepositions (*e.g.*, in, to, during) |
| ADV | Number of adverbs (*e.g.*, very, tomorrow, down) |
| AUX | Number of auxiliaries (*e.g.*, is, will do, should do) |
| CCONJ | Number of coordinating conjunctions (*e.g.*, and, or, but) |
| DET | Number of determiners (*e.g.*, a, an, the) |
| INTJ | Number of interjections (*e.g.*, psst, ouch, hello) |
| NOUN | Number of nouns (*e.g.*, girl, cat, air) |
| NUM | Number of numbers (*e.g.*, 1995, 7, seventy, XXII) |
| PART | Number of particles (*e.g.*, 's, not) |
| PRON | Number of pronouns (*e.g.*, I, she, you) |
| PROPN | Number of proper nouns (*e.g.*, James, USA, NATO) |
| PUNCT | Number of punctuations (*e.g..*, ?, !) |
| SCONJ | Number of subordinating conjunction (*e.g.*, if, while, that) |
| SYM | Number of symbols (*e.g.*, @, $, -) |
| VERB | Number of verbs (*e.g.*, runs, fly, eat) |
| X | Number of other type of constructs |
| SPACE | Number of spaces |
| STOPS | Number of stop words (*e.g.*, and, by, of) |
| NON_ALPHA | Number of non-alphabetic words (*e.g.*, 1, 1997, 33) |
| NAMED_ENTITIES | Number of named entities (London, Gary, EU) |

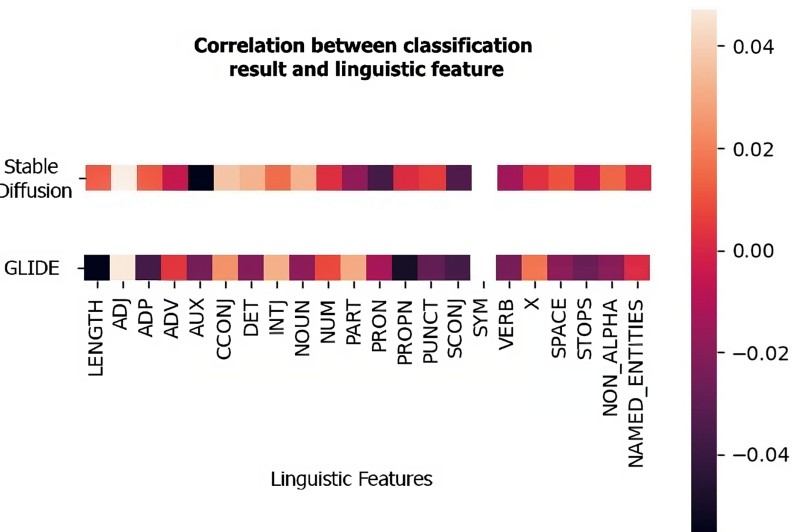

**Figure 4 The correlation between linguistic features and classification.** Pearson correlation values between the linguistic features and the classification provided by an MLP in the image+text setup (CLIP-ViT). The first row refers to the dataset composed of real images and images generated by Stable Diffusion while the second row refers to the dataset with GLIDE images.

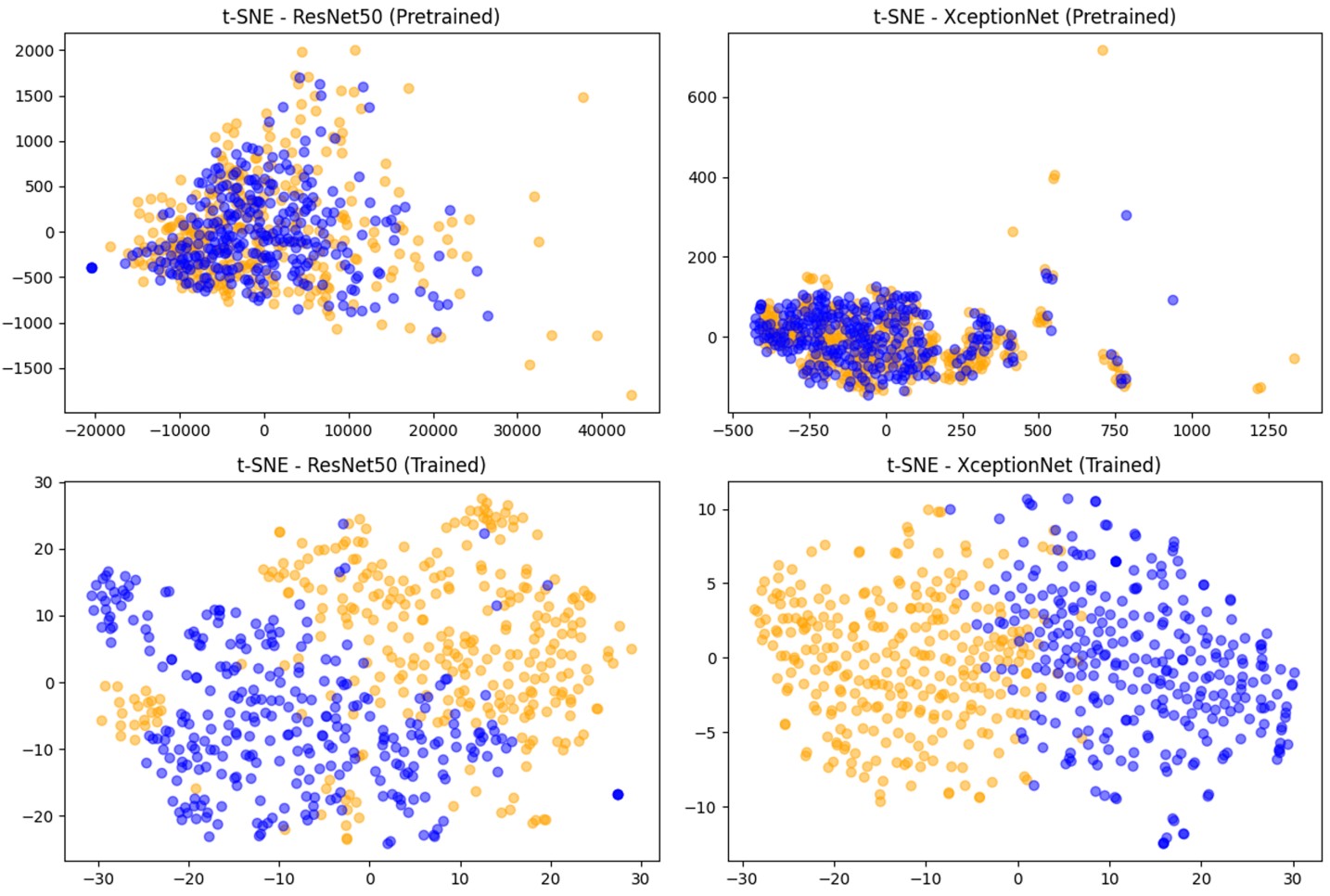

**Figure 5** **The features extracted using CNN-based methods.** t-SNE visualization of features extracted using Resnet50 or XceptionNet on 300 fake and 300 pristine images from MSCOCO dataset.

## Features exploration

In Fig. 5 we visualized the features extracted through Resnet50 and XceptionNet on 300 fake and 300 pristine images randomly selected from the MSCOCO validation set. These features were dimensionally reduced *via* PCA and visualized using t-SNE. As can be seen from the plots, the features extracted using only the pre-trained model of the two networks are not significant at all, causing an important overlap between pristine and fake classes. After training on the deepfake detection task, on the other hand, the features, especially of XceptionNet, are extremely good at discriminating between the two classes. Using these features allows the classification head of these models to easily achieve an incomparable level of performance since they already are very discriminative.

In Fig. 6 we show the same visualization but for the experiments where visual features are obtained *via* CLIP. It can be seen that in these cases, without any kind of fine-tuning on the task, the pristine and fake images features are somewhat more separated even if they still remain rather mixed. This is confirmed both in the image-only setup and in the case

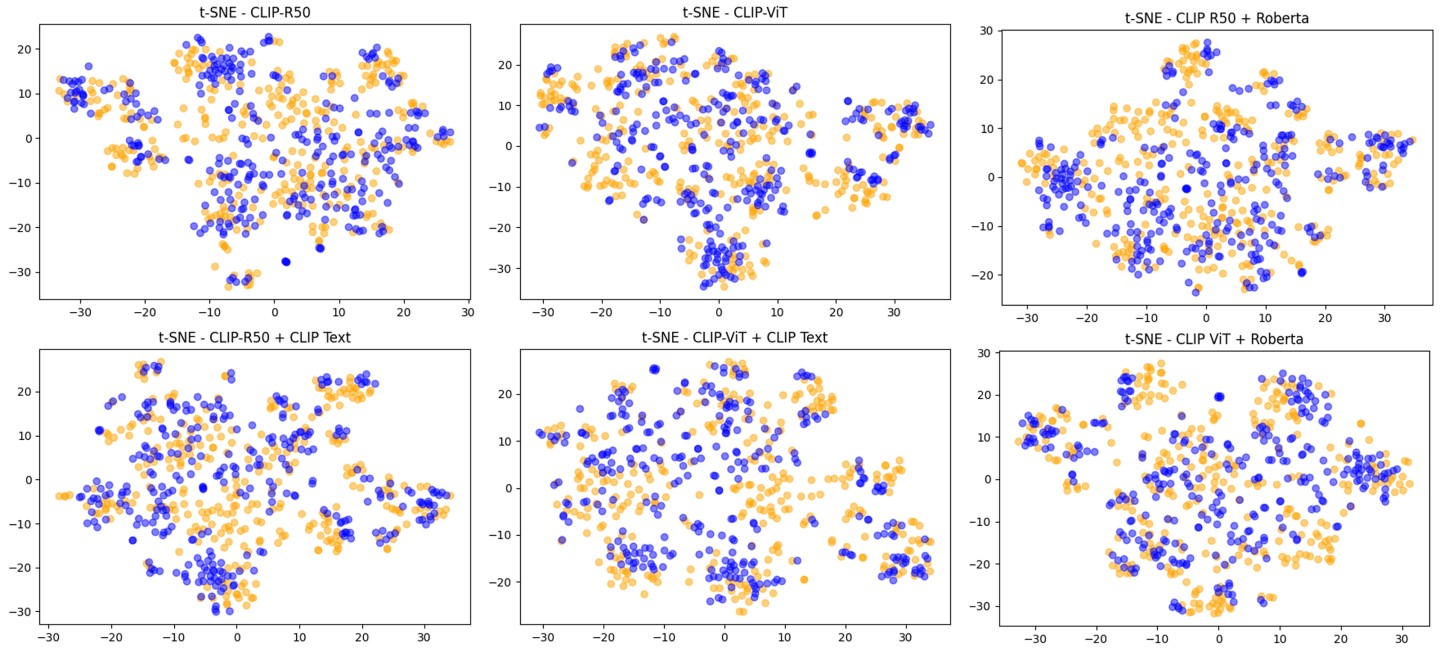

**Figure 6 The features extracted using CLIP-based methods.** t-SNE visualization of features extracted using CLIP for the visual part and CLIP or RoBERTa for the captions of 300 fake and 300 pristine images from the MSCOCO dataset.

where textual features (whether from CLIP or RoBERTa) are used. In this context, the MLP learns to discriminate between fake and pristine images by exploiting the peculiarities of these features without any fine-tuning of the backbones.

## CONCLUSIONS

In this study, we conducted some analysis on the detection of content generated by text-to-image systems, particularly Stable Diffusion and GLIDE. We tested several classifiers between MLPs and CNNs highlighting how classical deep learning models are easily able to distinguish images generated with these systems when they have seen examples of them in the training set, in particular when they conducted some pretraining on large datasets. By screening their generalization ability, however, they are rarely able to identify images generated by methods other than those used to construct the training set thus highlighting an important issue for these systems to be adopted in the real world. In that context, the introduction of textual features extracted from the images' captions can be helpful in some cases helping the detectors to identify inconsistencies between text and image. An analysis of the correlation between the credibility of the generated images and the category to which they belong as well as the composition of the caption associated with them was also conducted. From our experiments, the images generated by both the considered generators are more credible when they depict inanimate objects and thus result in greater error on the part of the classifiers. Conversely, images depicting people, animals, or animate subjects in general are easier to identify. In addition, there does not appear to be a strong correlation between the linguistic composition of the sentence and the classification ability

of the models. Some future works will include the exploration of different generators, datasets, and architectures and the development of new ideas to boost the generalization capabilities of the detectors. In this direction we could try to train models using a reduced set of data from multiple generators in an attempt to find out what amount and type of images generated by a given method are necessary to achieve optimal detection performance.

### Funding

This work was supported by the SERICS (PE00000014) and FAIR (PE00000013), funded under the MUR National Recovery and Resilience Plan funded by the European Union-NextGenerationEU, and by AI4Media (EC H2020 n. 951911). There was no additional external funding received for this study. The funders had no role in study design, data collection and analysis, decision to publish, or preparation of the manuscript.

### Grant Disclosures

The following grant information was disclosed by the authors:
SERICS: (PE00000014) and FAIR (PE00000013).
MUR National Recovery and Resilience Plan funded by the European Union—NextGenerationEU.
AI4Media: EC H2020 n. 951911.

### Competing Interests

Andrea Esuli is an Academic Editor for PeerJ CS.

### Author Contributions

- Davide Alessandro Coccomini conceived and designed the experiments, performed the experiments, analyzed the data, performed the computation work, prepared figures and/or tables, authored or reviewed drafts of the article, and approved the final draft.
- Andrea Esuli conceived and designed the experiments, analyzed the data, authored or reviewed drafts of the article, and approved the final draft.
- Fabrizio Falchi conceived and designed the experiments, authored or reviewed drafts of the article, and approved the final draft.
- Claudio Gennaro conceived and designed the experiments, authored or reviewed drafts of the article, and approved the final draft.
- Giuseppe Amato conceived and designed the experiments, authored or reviewed drafts of the article, and approved the final draft.

### Data Availability

   The code to generate our data is available at Zenodo:
   Davide Coccomini. (2024). davide-coccomini/Detecting-Images-Generated-by-Diffusers: Initial Release (v1.0). Zenodo. https://doi.org/10.5281/zenodo.10869925.

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
