# Peer review of "Detecting images generated by diffusers"

_PeerJ Computer Science, doi:10.7717/peerj-cs.2127_

## Round 0.1 · original submission · Major Revisions

With respect to the reviewers’ comments and my reading of the paper, the paper is overall well-structured and after suitable major revisions, could be reconsidered for publication in the journal. However, the reviewers have raised several issues and requested extra statistics. I have decided to give you the opportunity to respond to these issues and resubmit. Please respond to each of these issues point by point, making clear any changes to the manuscript. In particular, the reviewers have suggested revising the scope of the questions asked at the end of section one to establish novelty and differentiate your work from existing work on fake detectors. Please also respond to this suggestion. There are also some recommendations concerning improvements to section 3, the figures and tables for your consideration.
**Language Note:** PeerJ staff have identified that the English language needs to be improved. When you prepare your next revision, please either (i) have a colleague who is proficient in English and familiar with the subject matter review your manuscript, or (ii) contact a professional editing service to review your manuscript. PeerJ can provide language editing services - you can contact us at [email protected] for pricing (be sure to provide your manuscript number and title). – PeerJ Staff

Reviewer 1 ·

Basic reporting

1. Section 3 should be polished and enriched. In 3.1, I suggest the authors to describe the framework and implementation of the image-only setting first and then the image+text setting. And Figure 1 can also be cited here. In 3.2, the authors need to introduce the building method for the dataset more clearly, especially for the test set. Also the authors can use a table to summary the information of the datasets such as numbers, sources, generation method and so on. Besides, whether the captions used for generating are same as the captions of the real images should also be clarified. In 3.3, the structures of the models can be moved to Sec 3.1. And the authors should state whether the pretrained models are also tuned or frozen.

2. Please give the basis of the used features in Table 6. Related references may be needed. And relationship between the features and the ones extracted by CLIP or RoBERT should be discussed.

3. The authors should intruduce CLIP and RoBERT in related work and compare the differences between them.

4. Recently, many studies about detection of generated images are done. The authors can add them into Related Work and discuss their shortcomings.

5. In Table 1, the word "Caption" is used in Row 11 and Column 3 and it is used only once. Please confirm if there is any error.

6. Do the corresponding experiments in Table 1 and Table 3 use the same models, but only the testing images are different? And Table 2 and Table 4? The authors should clarify it.

Experimental design

1. Please add the experiments using un-pretrained XceptionNet and ResNet50 and analyse the effect of pretraining on a large dataset. If the pretrained models are frozen during training, is fine-tuning them beneficial?

2. The cross-method classification results on MLP-RoBERTa are needed.

3. Please provide True Positive Rate and True Negative Rate of each experiment. It can help understand the models' performance on real and fake images respectively.

4. Please visualize the real / fake image features extracted by XceptionNet, ResNet50, CLIP-R50 and CLIP-VIT, and the text features extracted by CLIP and RoBERTa. Discuss the distributation differences between two modailities, between real and fake images, and among different extractors. t-SNE may be used.

Validity of the findings

1. The authors say that there does not appear to be a strong correlation between the linguistic composition of the sentence and the classification ability of the models. However, the relationship between the features and the ones extracted by CLIP and RoBERT is not stated. The authors should further elaborate on the rationale of this experiment.

2. I believe that the research on the role of text in generated image detection is interesting. From the results, we see different outcomes brought by text. For example, when the model is trained using SD-generated data and CLIP-VIT, the accuracy on SD-generated images drops 1% and 3.1% for MLP-Base and MLP-RoBERT with the texts. But when the model is tested on GLIDE-generated images, the accuracy improves 4.8% with the texts. Please provide discussion about why text plays a positive or negative role under different conditions.

3. I think the 3rd problem proposed in Introduction (Line 54-56) is not addressed. From the results, we can conclude what model structure can get better detection results, what types of images are easier to classify, and whether the linguistic features are relevant to the task. But I think the authors do not identify "specific" features that can aid detection. The authors also do not summarize how to "develop more accurate and robust methods" in Abstract or Conclusion. Please reconsider it.

Cite this review as
Anonymous Reviewer (2024) Peer Review #1 of "Detecting images generated by diffusers (v0.1)". PeerJ Computer Science

Reviewer 2 ·

Basic reporting

Advantages:
+ Clear and unambiguous: The paper is written in fluent and professional English and is easy to follow.
+ Structure: The paper is mostly well-structured. Some of the images included in the figures are in low resolution, which should be ascribed to the inherent low-resolution results from GLIDE. Some of the figures can be better organized to visualize more information. For example, in Figure 3, more pairs of images can be included in the empty space without increasing the actual size of the figure.
+ Self-contained: The paper includes sufficient results to validate the proposed questions.
+ Theorems/Proofs: This paper is mainly about experimentally validating the DNN performance on detecting fake images from diffusion models, so there are no theoretical proofs involved. The involved terms, such as linguistic features, are explained clearly.

Drawback:
- Literature references: Literature studies are mostly satisfactory, including necessary discussions of image generative models, particularly diffusion models, as well as the methods to detect the synthetic images generated by these models. However, I think the paper should clearly distinguish itself from the existing works regarding fake detectors. For detailed reasons, please see the second part.

Experimental design

+ The research is original and satisfies the aims and scope of the journal.
+ Rigorous investigation: Detailed experiments are performed to investigate the ability of DNN to identify fake images from diffusion models.
+ Experiment methods and model designs are introduced with sufficient details. Plus, codes have been released for replication.

- Research questions are clearly defined and meaningful. However, it is questionable whether part of the raised research questions is already solved by some of the prior works, therefore I am not sure if some of the studied questions are true “knowledge gaps” that we are not clear. Specifically, the first question in Line 51 is “Can traditional deep learning models easily detect synthetic images obtained with text-to-image methods?”. This is an interesting question and has many discussions, such as in [1] (as cited in this paper), [2], [3]. My understanding is through these works, we have methods to detect these fake images, which answers this research question. For revision, authors might want to consider either stating why previous studies do not solve this question (which I think might be hard) or revising the scope of the questions, i.e. generalization problem, models for detection, etc.
[1] Corvi, Riccardo, Davide Cozzolino, Giada Zingarini, Giovanni Poggi, Koki Nagano, and Luisa Verdoliva. "On the detection of synthetic images generated by diffusion models." In ICASSP 2023-2023 IEEE International Conference on Acoustics, Speech and Signal Processing (ICASSP), pp. 1-5. IEEE, 2023.
[2] Ma, Ruipeng, Jinhao Duan, Fei Kong, Xiaoshuang Shi, and Kaidi Xu. "Exposing the fake: Effective diffusion-generated images detection." AdvML-Frontiers@ICML (2023)
[3] Bammey, Quentin. "Synthbuster: Towards detection of diffusion model generated images." IEEE Open Journal of Signal Processing (2023).

Validity of the findings

+ The underlying data are publicly available and are standard datasets used by many other research works in this field.
+ Conclusion is clear and linked to the original questions.

- From my point of view, the findings in this work do not clearly distinguish itself from some existing works, as described earlier in the earlier part.

Additional comments

I do think this paper somehow provides new knowledge, such as the generalization ability of the detectors and the performances of different architectures. However, my major concern is the first claimed question seems already solved by prior works. This paper does not clearly distinguish itself from these existing works in its current form, and therefore the new values brought by this paper seem blurred. Please consider revising the scope of the first question and add necessary discussions of the prior work accordingly.

Cite this review as
Anonymous Reviewer (2024) Peer Review #2 of "Detecting images generated by diffusers (v0.1)". PeerJ Computer Science

---

## Round 0.2 · Minor Revisions

The reviewers find the manuscript much improved and are satisfied that the revised manuscript addresses their concerns. In particular, the revision of the scope of the research questions to establish novelty and differentiate your work from existing work on fake detectors.

Please address minor comments relating to small grammar issues. Reviewer 2 mentions a possible additional experiment. I will leave this as an option. You may perform this experiment or discuss it as possible future work.

Reviewer 1 ·

Basic reporting

The caption of Sec 4.1 can be distinguishable from the one of 4.2, such as “in-method classification”.

Experimental design

I have no additional issue.

Validity of the findings

I have no additional issue.

Additional comments

I have no additional issue.

Cite this review as
Anonymous Reviewer (2024) Peer Review #1 of "Detecting images generated by diffusers (v0.2)". PeerJ Computer Science

Reviewer 2 ·

Basic reporting

The current manuscript demonstrates clear improvements in its reference to the literature, particularly better acknowledging prior studies on fake image detection. It builds upon these work by focusing on factors that influence detection performance, such as classifier architectures and image content.

I also concur with Reviewer 1 that the previous version needed a more detailed discussion of the experimental settings. In the current manuscript, these settings are explained more clearly and are easier to understand.

Consistent to my prior review, I believe this manuscript is self-contained, including sufficient results to validate the proposed questions. I have no further concerns regarding the standards of the basic reporting section.

Experimental design

The modified research questions distinguish themselves from prior works. The current research questions are well-defined and present an interesting area for study.

I don’t have concerns about the current experimental design. However, after changing the research questions, given that one of the major focuses is to study the generalization problem in detecting fake images, I believe the current experiment could be enriched to better reveal this issue. Specifically, the current experiments clearly show that (a) when trained with fake images from Model A and tested on the same model, the classification model mostly succeeds; (b) when trained with fake images from Model A and tested on images from Model B, the classification model often fails. Additionally, it might be insightful to explore (c) how the model performs on Model B when trained with X% of fake images from Model A and Y% from Model B. We know that when Y=0, the performance is low. Would the performance increase significantly even if Y is small (say, Y=10%)? If so, that shows the generalization problem could potentially be solved by few-shot fine-tuning.

That said, I believe the current experimental design in the manuscript is mostly sufficient to address the raised research questions. The suggested experiment could also serve as a topic for future work discussions.

Validity of the findings

This work provides sufficient details for replication, including original code in the repository, detailed experiment and hyperparameter settings, underlying data, etc. The conclusion is clear and supported by experiment results.

Additional comments

There are some minor grammar issues in the paper:
* Citation format: In lines 151 and 315, the manuscript uses brackets with the author’s name as a reference, whereas overall this manuscript generally uses parentheses with numbers, e.g., (1). Are there any specific reasons for this variation in citation format?
* Grammar issues: There is a double period in line 208.

Cite this review as
Anonymous Reviewer (2024) Peer Review #2 of "Detecting images generated by diffusers (v0.2)". PeerJ Computer Science

---

## Round 0.3 · accepted · Accept

I have assessed the revised manuscript and I am satisfied that all the minor revisions raised by the reviewers have now been addressed. I am pleased to say that this manuscript is ready for publication.